# Artificial Intelligence for Risk Prediction of End-Stage Renal Disease in Sepsis Survivors with Chronic Kidney Disease

**DOI:** 10.3390/biomedicines10030546

**Published:** 2022-02-24

**Authors:** Kuo-Hua Lee, Yuan-Chia Chu, Ming-Tsun Tsai, Wei-Cheng Tseng, Yao-Ping Lin, Shuo-Ming Ou, Der-Cherng Tarng

**Affiliations:** 1Division of Nephrology, Department of Medicine, Taipei Veterans General Hospital, Taipei 11217, Taiwan; khlee5@vghtpe.gov.tw (K.-H.L.); mttsai5@vghtpe.gov.tw (M.-T.T.); wctseng@vghtpe.gov.tw (W.-C.T.); linyp@vghtpe.gov.tw (Y.-P.L.); 2Faculty of Medicine, School of Medicine, National Yang Ming Chiao Tung University, Taipei 11217, Taiwan; 3Institute of Clinical Medicine, National Yang Ming Chiao Tung University, Taipei 11217, Taiwan; 4Center for Intelligent Drug Systems and Smart Bio-Devices (IDS2B), Hsinchu 30010, Taiwan; 5Information Management Office, Taipei Veterans General Hospital, Taipei 11217, Taiwan; ycchu5@vghtpe.gov.tw; 6Big Data Center, Taipei Veterans General Hospital, Taipei 11217, Taiwan; 7Department and Institute of Physiology, National Yang-Ming University, Taipei 11217, Taiwan

**Keywords:** sepsis, chronic kidney disease, machine learning, artificial intelligence, end-stage renal disease

## Abstract

Sepsis may lead to kidney function decline in patients with chronic kidney disease (CKD), and the deleterious effect may persist in patients who survive sepsis. We used a machine learning approach to predict the risk of end-stage renal disease (ESRD) in sepsis survivors. A total of 11,661 sepsis survivors were identified from a single-center database of 112,628 CKD patients between 2010 and 2018. During a median follow-up of 3.5 years, a total of 1366 (11.7%) sepsis survivors developed ESRD after hospital discharge. We adopted the random forest, extra trees, extreme gradient boosting, light gradient boosting machine (LGBM), and gradient boosting decision tree (GBDT) algorithms to predict the risk of ESRD development among these patients. GBDT yielded the highest area under the receiver operating characteristic curve of 0.879, followed by LGBM (0.868), and extra trees (0.865). The GBDT model revealed the strong effect of estimated glomerular filtration rates <25 mL/min/1.73 m^2^ at discharge in predicting ESRD development. In addition, hemoglobin and proteinuria were also essential predictors. Based on a large-scale dataset, we established a machine learning model computing the risk for ESRD occurrence among sepsis survivors with CKD. External validation is required to evaluate the generalizability of this model.

## 1. Introduction

Sepsis refers to a syndrome of physiological dysregulation caused by infection. Without prompt treatment, the syndrome may lead to life-threatening organ failure. According to population-level epidemiologic data, 31.5 million patients experience sepsis worldwide each year, with a mortality rate as high as 16.8% [1]. Studies have shown that approximately 55% of septic patients have severe underlying disorders [2,3]. Chronic kidney disease (CKD) is a critical risk factor for severe sepsis because of the associated complications, such as malnutrition, endothelial dysfunction, and immunodeficiency [4,5]. Moreover, even after receiving treatment for infection, septic patients exhibit a high mortality rate and risk of renal function deterioration in the long term [6,7]. Therefore, clinical practice guidelines recommend regular follow-up of adult survivors of sepsis or septic shock, particularly those with preexisting kidney disease [8].

Sepsis is the predominant cause of hospital-acquired acute kidney injury (AKI) [9]. In one large multicenter study involving 120,123 critically ill patients, 27.8% (~30,000) had a primary diagnosis of sepsis; among them, the incidence of AKI was 42.1% [10]. Sepsis causes hemodynamic instability, leading to ischemic nephropathy. Cytokines have been shown to aggravate endothelial dysfunction and subsequent microvascular inflammation [11]. Moreover, patients with sepsis-associated AKI had a higher risk of CKD and end-stage renal disease (ESRD) than nonseptic patients with AKI [12,13]. Studies have shown the diagnostic and prognostic value of novel biomarkers other than creatinine for AKI [14,15]. However, the application of these biomarkers is limited by cost and low accuracy when confounders—such as old age, severe inflammation, advanced kidney disease, or liver cirrhosis—are not adjusted [16]. The application of artificial intelligence (AI) in the early prediction and risk stratification of AKI has attracted considerable research attention. Previous studies have demonstrated the use of machine learning algorithms for the early prediction of AKI 48 h after intensive care unit (ICU) admission and for distinguishing between transient and persistent AKI in patients following sepsis or cardiac surgery [17,18,19]. Nevertheless, these algorithms could not predict subsequent ESRD development. To this end, we used a large-scale CKD cohort and machine learning methods to develop a prediction model for renal function decline after sepsis to identify sepsis survivors at a high risk of CKD progression and ESRD.

## 2. Materials and Methods

### 2.1. Study Design and Data Source

We established a cohort of sepsis survivors based on data extracted from the Taipei Veterans General Hospital (VGH) Big Data Center; the data included detailed patient demographic and clinical information, diagnostic/procedural information, drug prescriptions, procedural codes, and laboratory data from 2010 to 2018. By using diagnostic codes from the International Classification of Diseases, Ninth and Tenth Revision (ICD-9-CM and ICD-10-CM, respectively), we identified 112,628 CKD patients (*ICD-9*: 585 and *ICD-10*: N18) aged ≥20 years from the database. We collected the data of individuals discharged alive with diagnostic codes involving sepsis (*ICD* codes 038, 995.91, A40, and A41), severe sepsis (*ICD* codes 995.92 and R65.20), and septic shock (*ICD* codes 785.52 and R65.21). We excluded patients with estimated glomerular filtration rate (eGFR) <15 mL/min/1.73 m^2^ at discharge, those requiring maintenance dialysis or kidney transplantation, and those younger than 20 years of age. We also excluded those with less than two serum creatinine measurement values to assess the decline in eGFR. Finally, we enrolled 11,661 eligible sepsis survivors with a history of CKD (Figure 1). Our study protocol fulfilled the ethical guidelines of the Declaration of Helsinki and was approved by the Institutional Review Board of Taipei VGH (2021-03-012AC).

### 2.2. Input Features

Input features comprised the demographic characteristics of age, sex, smoking status, and alcohol consumption; the presence of the comorbidities of hypertension (HTN), diabetes mellitus (DM), coronary artery disease (CAD), congestive heart failure, peptic ulcer disease, chronic obstructive pulmonary disease, malignancy; Charlson Comorbidity Index (CCI) scores; laboratory data, including hemoglobin (HGB), total cholesterol, glycohemoglobin, eGFR, and the spot urine protein/creatinine ratio (UPCR); and medication prescriptions at discharge, including calcium channel blockers (CCBs), beta-blockers, alpha-blockers, renin–angiotensin system (RAS) inhibitors, antiplatelets, warfarin, statins, diuretics, nonsteroidal anti-inflammatory drugs (NSAIDs)/cyclooxygenase (COX)-2 inhibitors, miscellaneous oral hypoglycemic agents, and insulin. The eGFR was determined using the chronic kidney disease epidemiology collaboration (CKD-EPI) creatinine equation.

### 2.3. Outcomes and Class Definition

The endpoint was ESRD development defined by an eGFR < 15 mL/min/1.73 m^2^ and initiation of long-term hemodialysis/peritoneal dialysis or kidney transplantation during the follow-up period. CKD patients were followed up until death or the end of the study period. In the machine learning algorithm, we annotated the class as 1 for sepsis survivors who developed ESRD; otherwise, the class was 0 if no event occurred.

### 2.4. Construction of Machine Learning Models

In data preprocessing, we imputed the missing values of the variables by the k nearest neighbors algorithm [20]. The data were randomly divided into a training dataset and a validation dataset with a ratio of 70:30. We adopted various machine learning algorithms to construct the models, namely logistic regression, random forest, extra trees, extreme gradient boosting (XGBoost), light gradient boosting machine (LGBM), and gradient boosting decision tree (GBDT), to predict ESRD after surviving sepsis. We examined the predictive power based on the area under the receiver operating characteristics curve (AUC) and the precision–recall curve to optimize the models with the best performance. The machine learning model with the highest AUC was then compared with the kidney failure risk equation (KFRE), predicting the 2- and 5-year risk of progression to ESRD among the sepsis survivors [21]. The variables in KFRE include age, sex, eGFR, and albuminuria, defined by urine albumin/creatinine ratio (UACR). Using a validated conversion formula, we converted the UPCR to UACR for use in KFRE equation [22]. We used Shapley additive explanation (SHAP) values to evaluate the importance of each input features contributing to the model. Statistical analysis and mapping were performed using Python (version 3.7.6, available at http://www.python.org. Accessed 16 February 2022).

## 3. Results

### 3.1. Study Population

A total of 11,661 sepsis survivors from 2010 to 2018 were included in our final cohort, and their clinical features are presented in Table 1. The participants were predominantly male and aged around 75 years, and 30% had a smoking and alcohol consumption history. The CCI score was 4, and 64.7% of the patients had underlying HTN, 51.8% had DM, and 30.7% had CAD. Regarding renal function, the patients had a median baseline creatinine level of 1.1 mg/dL and an eGFR of 59.3 mL/min/1.73 m^2^ at hospital discharge. CCBs (55.0%) were the most commonly used concomitant medications, followed by angiotensin-converting enzyme inhibitors and angiotensin II receptor antagonists (49.0%). Although sodium/glucose cotransporter 2 (SGLT2) inhibitors have been proven beneficial in the deceleration of CKD progression, and less than 1% of patients were prescribed this class of drugs. Genitourinary and respiratory infections were the two most common etiologies of sepsis in this cohort. During the follow-up period, we recorded 2573 (22.1%) death, 3031 (26.0%) patients encountered re-hospitalization, and 1366 (11.7%) of them developed ESRD. We divided the sepsis survivors randomly into 2 groups and allocated 70% of them to the training set and the remaining 30% to the validation set.

### 3.2. Model Performance for Predicting End-Stage Renal Disease Development

To predict the risk of ESRD in sepsis survivors by using AI, we adopted various machine learning models based on various clinical features at discharge obtained from the index hospital. The GBDT model had the highest AUC of 0.879, and the LGBM model had the second-highest AUC of 0.868 (Table 2). The extra trees and random forest algorithms had AUCs of 0.865 and 0.864, respectively. Compared with the LGBM model, the GBDT model had higher accuracy, F1 score, precision, and recall. The sensitivity and specificity between the two models were comparable. The receiver operating characteristic curves and precision–recall curves of the models are shown in Figure 2.

### 3.3. Feature Importance

Figure 3a shows the top 25 clinical features predicting ESRD by the GBDT model. According to the mean SHAP values of each feature, we sorted the impacts from high to low in descending order. In this model, the top 5 variables were eGFR, HGB, UPCR, insulin, and β-blockers use. In Figure 3b, the SHAP summary plot illustrated the impact of clinical features on model output. A positive SHAP value for a feature means that the value of that feature contributes positively to the prediction. The higher the SHAP value of a feature, the higher the probability of positive output of this model. A dot is represented for each feature value of each patient, and the color represents the feature value (high in red, low in blue). According to our prediction model, the low eGFR showed the most significant impact on ESRD prediction.

As shown in Figure 4a–c, the SHAP dependence plots depicted the effect of individual features on the ESRD risk prediction in the GBDT model. The values on the *y*-axis indicated the SHAP values of features, and values on the *x*-axis were eGFR in mL/min/1.73 m^2^, HGB in g/dL, and UPCR in mg/mg, respectively. The impact of eGFR on predicting ESRD increased while eGFR values < 25 mL/min/1.73 m^2^ at hospital discharge. We also found that HGB levels < 10 g/dL and UPCR > 2 mg/mg improved ESRD risk prediction.

Figure 4d–f illustrated the interaction between the SHAP values of eGFR and the use of insulin, RAS inhibitors, and HTN. These variables are positively correlated with the predictive value of eGFR for future ESRD, particularly in sepsis survivors with substantially low eGFR levels at hospital discharge.

### 3.4. Performance of Machine Learning Model Versus Kidney Failure Risk Equation

In Figure 5, we compared the performance of the GBDT model with KFRE by using the area under the receiver operating characteristic (ROC) curves. In our study cohort, the AUCs assessing the 2-year ESRD risk for the GBDT model and KFPE were 0.886 and 0.857, whereas the AUCs assessing the 5-year ESRD risk were 0.879 and 0.848, respectively. The machine learning model not only had higher AUCs, but also had better accuracy and average precision than KFPE. The sensitivity and specificity between the two models were similar.

## 4. Discussion

In this retrospective cohort study of 11,661 sepsis survivors with CKD, during a median follow-up of 3.5 years, approximately 11.7% of the participants developed ESRD after discharge. We established machine learning models to predict the risk of ESRD in sepsis survivors. The GBDT model had the highest predictive performance with the highest AUC of 0.879 among all the algorithms. Among the clinical features at hospital discharge, the eGFR < 25 mL/min/1.73 m^2^ was determined to be the most critical predictive factor for ESRD, followed by HGB < 10 g/dL and UPCR > 2mg/mg. The use of RAS inhibitors and the presence of HTN were positively correlated with the predictive value of eGFR for future ESRD. Our results also showed that machine learning models trained with the best performing clinical variables might predict the ESRD development more accurately than a conventional risk scoring system.

Sepsis is a critical cause of death in CKD patients. In severe sepsis and septic shock, tissue hypoxia and a series of cellular derangements (e.g., lactic acid production and microcirculatory and mitochondrial dysfunction) may occur, which may lead to AKI and even multiple-organ failure [23]. Moreover, even if a patient survives after prompt treatment, the risk of CKD progression or re-hospitalization after discharge remains high. Joana et al. conducted a retrospective cohort study of 256 critically ill patients with sepsis-associated AKI to evaluate their risk of adverse renal outcomes after surviving sepsis. Post-discharge 30-day and long-term mortality rates were 21.4% and 44.1%, respectively, whereas the percentage of patients requiring long-term dialysis was 16.5% during follow-up [24]. Moreover, Hallie et al. indicated that survivors of severe sepsis had a higher re-hospitalization rate of 42.7% for sepsis relapse and AKI compared with comorbidity-matched patients without sepsis [25]. Our results revealed lower mortality of 22.1% and re-hospitalization rates of 26% in sepsis survivors than those in previous studies, which might be attributed to differences in disease severity. Moreover, we included all inpatients with sepsis, whereas the study population in previous studies was mainly composed of critically ill patients in ICUs. However, the mean eGFR value and incidence of ESRD after discharge in patients who survived sepsis in our study and in studies evaluating the long-term prognosis after sepsis are similar.

In the present study, the median eGFR level was 59.3 mL/min/1.73 m^2^ among the sepsis survivors with CKD, categorized as stage 3a according to The Kidney Disease: Improving Global Outcomes (KDIGO) 2012 Clinical Practice Guideline. Of the patients, 1366 (11.7%) progressed to ESRD during a median follow-up of 3.5 years. We used clinical indicators to predict the future risk of ESRD by using machine learning models. In previous studies that used AI to predict the progression of renal function, the primary endpoints were the occurrence of AKI or acute kidney disease following ICU admission, cardiovascular surgery, or sepsis [17,18,19]. Nevertheless, the follow-up period of 1–2 years in these studies was too short to effectively assess the long-term eGFR decline or progression to ESRD. Moreover, information on the impact of sepsis on the risk of renal adverse outcomes is lacking, particularly for patients with CKD. The present study is the first to establish machine learning models for predicting the risk of ESRD among CKD patients who survive sepsis, and the GBDT algorithm yielded an accuracy as high as 0.879, as measured using the AUC. Our study cohort revealed that the GBDT model had better performance metrics than KFRE, created by Tangri et al., demonstrating the high accuracy of risk equations predicting CKD progression to kidney failure using age, sex eGFR, and UACR in patients with CKD stage 3–5 [21,26]. The better predictive power of the GBDT model might be attributed to the discrepancy in the original study population, differences among subjects in CKD severity, and the complete inclusion of variables and data training in the machine learning approach. However, our model still lacks external validation to demonstrate its predictive power among nonseptic patients and other CKD subpopulations.

The machine learning model revealed the strong effect of eGFR < 25 mL/min/1.73 m^2^ at discharge in predicting ESRD, with a positive SHAP value around 5. This finding is consistent with those of previous epidemiological studies and systematic reviews that investigated the baseline variables associated with subsequent renal outcomes among individuals who survive AKI [27,28,29]. According to the SHAP dependence plot of eGFR with RAS inhibitors in the ESRD model, the apparent interaction with RAS inhibitors increases the influence of eGFR in predicting ESRD development. Supraphysiological GFR levels were associated with fewer RAS inhibitors and lower SHAP values. Our results may reflect that persistent glomerular hyperfiltration sometimes leads to out-of-model predicted GFR declines and poor long-term renal outcomes. Our model also suggested that HGB level is a crucial predictive factor for ESRD risk after sepsis. The recommended optimal HGB level for patients with CKD is >10 g/dL, and the level observed in our cohort was 10.5 g/dL. Although the current evidence does not confirm that maintaining an appropriate HGB level in patients with CKD can decelerate the deterioration of renal function, the SHAP plot of our AI models demonstrated that a higher HGB level in sepsis survivors was associated with a lower risk of ESRD. Because inflammation and systemic infection suppress erythropoiesis in the bone marrow, this finding may suggest the effectiveness of erythropoietin administration concomitant with antimicrobial therapy to maintain appropriate HGB levels in patients with CKD who experience sepsis during admission.

As proteinuria is a known biomarker for CKD progression, some reports have highlighted urine protein as a diagnostic predictor of AKI in various cases of nephrotoxicity induced by drugs, including cisplatin and NSAIDs. In a large cohort of nearly 1 million adults, James et al. demonstrated an independent association among eGFR, proteinuria, and incidence of AKI [30]. They reported that patients with eGFR levels of ≥60 mL/min/1.73 m^2^ and mild proteinuria (urine dipstick, trace to 1+) had a 2.5 times higher risk of admission to a hospital for AKI than patients without proteinuria. The risk was increased by 4.4-fold in those with severe proteinuria (urine dipstick ≥ 2+). Adjusted rates of hospitalization and dialysis for AKI remained high in patients with heavy dipstick proteinuria, irrespective of eGFR level. In the assessment, serial evaluation, and subsequent sequelae of acute kidney injury (ASSESS-AKI) study, in the matched cohort of 1538 participants, half of whom had AKI during hospitalization, higher urine albumin/creatinine ratio quantified 3 months after hospital discharge was associated with an increased risk of kidney disease progression and served as a risk discriminator [31]. These findings confirm the suggestion of previous reports that proteinuria is a potent risk factor for subsequent AKI and CKD progression. In addition to the RAS inhibitors, recent trials have revealed the benefit of SGLT2 inhibitors in reducing proteinuria and ESRD risk in both diabetic and nondiabetic CKD. In the “Empagliflozin Cardiovascular Outcome Event Trial in Type 2 Diabetes Mellitus Patients—Removing Excess Glucose” (EMPA-REG OUTCOME) trial, empagliflozin markedly reduced the risk of major adverse cardiovascular events and delayed CKD progression in patients with concomitant type 2 DM and cardiovascular disease [32]. In the “Dapagliflozin and Prevention of Adverse Outcomes in Chronic Kidney Disease” (DAPA-CKD) trial, the use of dapagliflozin 10 mg once daily resulted in a 39% reduction in the risk of declining kidney function, the onset of ESRD, or kidney failure death among patients with an eGFR of 25–75 mL/min/1.73 m^2^, irrespective of diabetes status [33]. Because we established this cohort on the basis of medical records before 2018, less than 1% of patients were reported to have used SGLT2 inhibitors. However, according to the KDIGO 2020 guideline recommendation of using SGLT2 inhibitors in managing patients with T2DM, CKD, and those with an eGFR ≥ 30 mL/min/1.73 m^2^, and extending the indications for an SGLT2 inhibitor in nondiabetic CKD patients, recognition of the importance of SGLT2 inhibitors in the CKD prediction model has been increasing [34]. Therefore, further investigation of this topic is imperative.

Our study has several strengths. This study was the first to use machine learning algorithms for predicting long-term renal outcomes after discharge based on a large-scale CKD data set that integrates all records from hospitalization to discharge and after discharge. In addition, we included sepsis survivors who had at least two serum creatinine measurements. Therefore, we could define renal endpoints for sepsis survivors stratified by eGFR level rather than diagnostic codes as in other studies extracted data from administrative datasets.

Nevertheless, this study has several limitations. First, this was a single-center study, and external validation is needed to evaluate the generalizability of the prediction models in this study. Second, we defined the sepsis survivors on the basis of *ICD* codes from the data set. Bias due to hospital discharge coding may potentially affect the risk prediction of regression and AI models. Third, although we indicated the presence of HTN was positively correlated with the predictive value of eGFR for future ESRD, we mainly defined HTN by diagnostic codes and anti-HTN medication prescriptions. Hence, the predictive value of serial blood pressure changes for ESRD risk warrants further investigation. Moreover, the biomarkers of tubular injury, such as cystatin C [35], kidney injury molecule-1 [36], liver fatty acid-binding protein [37], and neutrophil-gelatinase-associated lipocalin [38]—which provide potential diagnostic information but are not routinely examined in clinical settings—were absent from the models developed in the present study. Finally, our data did not account for the protective role of SGLT2 inhibitors in sepsis survivors. Nevertheless, based on the evidence of kidney and cardiovascular protection offered by SGLT2 inhibitors in CKD patients, our prediction model may assist physicians in decision making regarding recommending this drug for sepsis survivors who are at a high risk of ERSD.

## 5. Conclusions

Our study established a machine learning model revealing that sepsis survivors with an eGFR < 25 mL/min/1.73 m^2^, HGB < 10 g/dL, and UPCR > 2 mg/mg at hospital discharge had a higher risk of progression from CKD to ESRD. Our cohort also revealed that this AI-based predictive model outperformed conventional scoring systems predicting ESRD. External validation is warranted to prove the generalizability of this model, which may provide an early warning and may help improve the prognosis of sepsis survivors at high risk of developing ESRD.

## Figures and Tables

**Figure 1 biomedicines-10-00546-f001:**
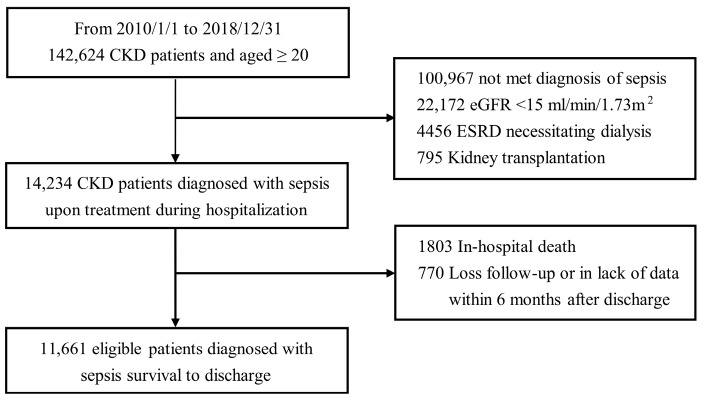
Study flowchart. This retrospective cohort consisted of 142,624 adults who received a diagnosis of chronic kidney disease between 2010 and 2018—a total of 14,234 patients with concomitant sepsis were treated in the hospital. After excluding patients who died during admission and those with insufficient data, we finally enrolled 11,661 patients who survived sepsis to discharge. Abbreviations: CKD—chronic kidney disease; eGFR—estimated glomerular filtration rate; ESRD—end-stage renal disease.

**Figure 2 biomedicines-10-00546-f002:**
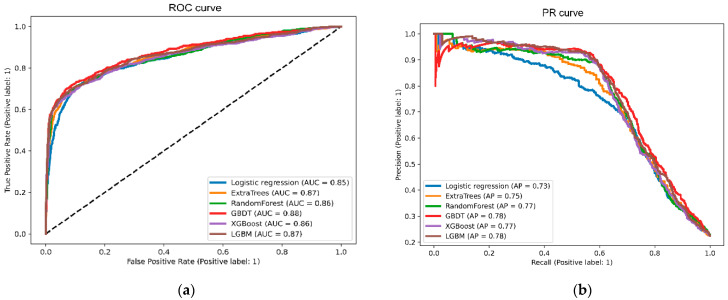
Receiver operating characteristic curves (**a**) and precision–recall curves (**b**) of machine learning models. GBDT yielded the highest area under the receiver operating characteristic curve and average precision, followed by LGBM and extra trees. Abbreviations: ROC—receiver operating characteristic; AUC—area under the curve; GBDT—gradient boosting decision tree; XGBoost—extreme gradient boosting; LGBM—light gradient boosting machine; PR—precision–recall; AP—average precision.

**Figure 3 biomedicines-10-00546-f003:**
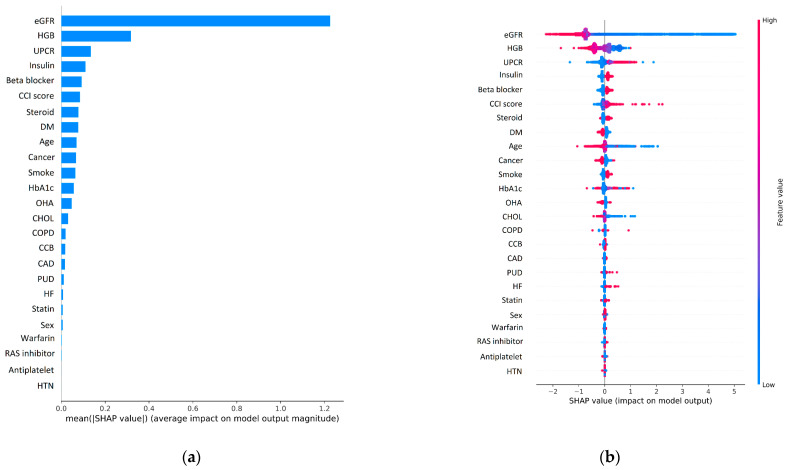
Feature importance plot (**a**) and SHAP summary plot (**b**) of the top 25 clinical features predict end-stage renal disease development in the GBDT model. Abbreviations: eGFR—estimated glomerular filtration rates; HGB—hemoglobin; UPCR—urine protein/creatinine ratio; CCI—Charlson comorbidity index; DM—diabetes mellitus; HbA_1C_—glycohemoglobin; OHA—oral hypoglycemic agents; CHOL—cholesterol; COPD—chronic obstructive pulmonary disease; CCB—calcium channel blockers; CAD—coronary artery disease; PUD—peptic ulcer disease; HF—heart failure; RAS—renin–angiotensin system; HTN—hypertension.

**Figure 4 biomedicines-10-00546-f004:**
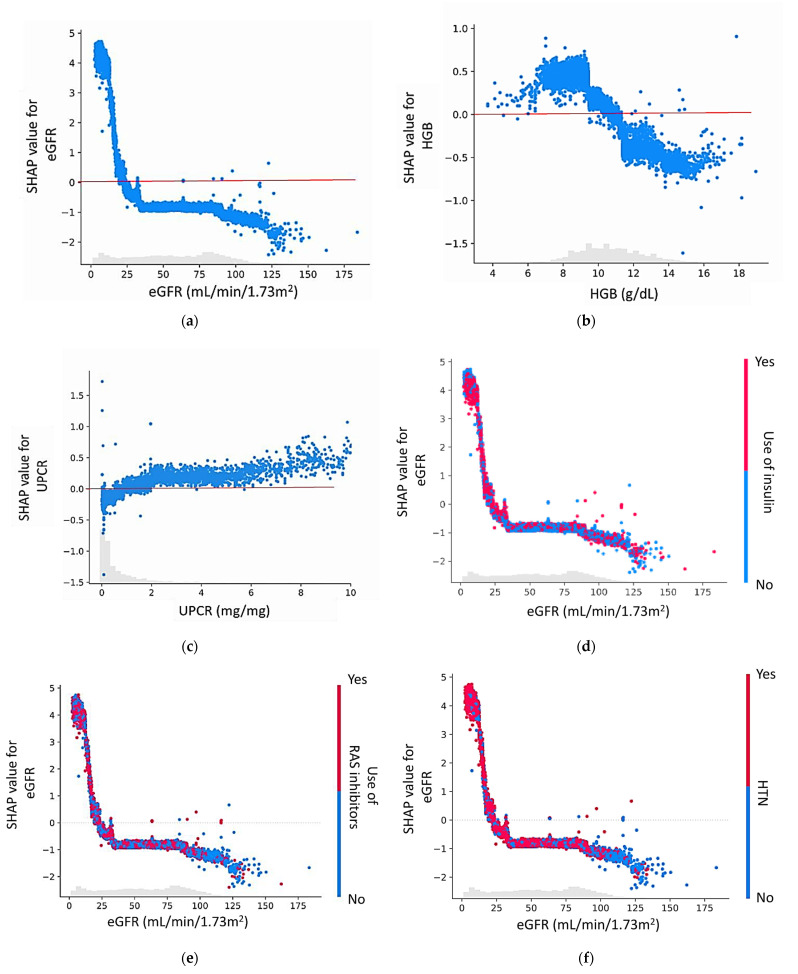
SHAP dependence plots of the GBDT model. Figure 4a–c showed the impact of eGFR (**a**), HGB (**b**), and UPCR (**c**) on the prediction model’s output. The risk prediction for end-stage renal disease development increases while the SHAP values of specific features exceed zero, represented by the red lines. Figure 4d–f showed the interaction effects between eGFR and the use of insulin (**d**), eGFR and the use of RAS inhibitor (**e**), eGFR and HTN (**f**) in the prediction model. The dotted lines represent while the SHAP value is zero. Abbreviations: SHAP—Shapley additive explanation; eGFR—estimated glomerular filtration rates; HGB—hemoglobin; UPCR—urine protein/creatinine ratio; RAS—renin–angiotensin system; HTN—hypertension; GBDT—gradient boosting decision tree. Red/blue—Features that push the prediction higher are shown in red, and those pushing the prediction lower are blue. A gray area refers to the patient distribution by eGFR levels.

**Figure 5 biomedicines-10-00546-f005:**
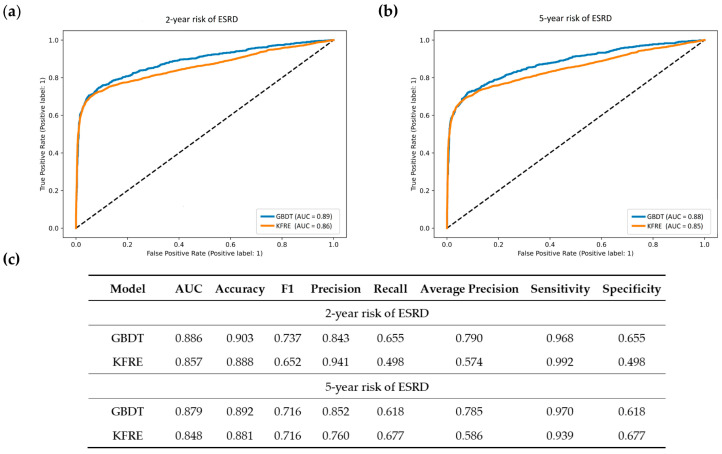
Receiver operating characteristic curves of GBDT prediction model and kidney failure risk equation (KFPE) predicting 2-year risk of ESRD (**a**) and 5-year risk of ESRD (**b**). Comparing the prediction performance of the two models, the GBDT model had a higher AUC, accuracy, and average precision than KFPE in predicting 2-year and 5-year risk of ESRD (**c**). The diagonal dotted line represents an AUC of 0.5. Abbreviations: ROC—receiver operating characteristic; AUC—area under curve; GBDT—gradient boosting decision tree; KFRE—kidney failure risk equation; ESRD—end-stage renal disease.

**Table 1 biomedicines-10-00546-t001:** Clinical features of the patients in the training and validation sets used in machine learning models.

	All	Training Set	Validation Set
	(*n* = 11,661)	(*n* = 8162)	(*n* = 3499)
Demographic and Clinical Characteristics			
Age, years	76.7 (63.3, 85.5)	76.7 (63.3, 85.5)	76.7 (63.1, 85.6)
Male sex, *n* (%)	6927 (59.4)	4865 (59.6)	2062 (58.9)
Smoking, *n* (%)	4289 (36.8)	3009 (36.9)	1280 (36.6)
Alcohol consumption, *n* (%)	3291 (28.2)	2318 (28.4)	973 (27.8)
ICU admission, *n* (%)	6367 (54.6)	4457 (54.6)	1910 (54.6)
Use of mechanical ventilators, *n* (%)	4291 (36.8)	3004 (36.8)	1287 (36.8)
Use of inotropic agents, *n* (%)	5562 (47.7)	3893 (47.7)	1669 (47.7)
Underlying Comorbidities			
Hypertension, *n* (%)	7540 (64.7)	5270 (64.6)	2270 (64.9)
Diabetes mellitus, *n* (%)	6046 (51.8)	4234 (51.9)	1812 (51.8)
Coronary artery disease, *n* (%)	3576 (30.7)	2511 (30.8)	1065 (30.4)
Heart failure, *n* (%)	2551 (21.9)	1792 (22.0)	759 (21.7)
Peptic ulcer disease, *n* (%)	2822 (24.2)	1990 (24.4)	832 (23.8)
COPD, *n* (%)	2267 (19.4)	1606 (19.7)	661 (18.9)
Malignancy, *n* (%)	4886 (41.9)	3422 (41.9)	1464 (41.8)
Charlson comorbidity index	4 (3, 6)	4 (3, 6)	4 (2, 6)
Laboratory Data at Hospital Discharge			
White blood cells,/mm^3^	8100 (5700, 11,900)	8100 (5700, 11,900)	8100 (5700, 12,000)
HGB, g/dL	10.5 (9.3, 12.0)	10.5 (9.3, 12.0)	10.5 (9.3, 12.0)
Total cholesterol, mg/dL	160.0 (134.0, 188.0)	160.0 (134.0, 189.0)	159.0 (133.0, 187.0)
LDL-C, mg/dL	91.0 (70.0, 114.0)	91.0 (70.0, 115.0)	91.0 (69.0, 113.0)
HDL-C, mg/dL	41.0 (32.0, 51.0)	41.0 (32.0, 51.0)	41.0 (32.0, 51.0)
Glucose, mg/dL	116.0 (95.0, 156.0)	116.0 (94.0, 155.0)	117.0 (95.0, 157.0)
Uric acid, mg/dL	5.5 (4.1, 7.1)	5.5 (4.1, 7.1)	5.6 (4.1, 7.1)
HbA_1c_, %	7.2 (6.1, 10.3)	7.1 (6.1, 10.3)	7.2 (6.1, 10.5)
Albumin, mg/dL	3.0 (2.6, 3.4)	3.0 (2.6, 3.4)	3.0 (2.6, 3.4)
Blood urea nitrogen, mg/dL	24.0 (14.0, 51.0)	24.0 (14.0, 51.0)	24.0 (14.0, 50.0)
Creatinine, mg/dL	1.1 (0.7, 2.1)	1.1 (0.7, 2.2)	1.1 (0.7, 2.1)
eGFR, mL/min/1.73 m^2^ *	59.3 (35.5, 83.6)	59.2 (33.4, 83.6)	59.3 (35.1, 83.2)
C-reactive protein, mg/dL	3.4 (1.2, 9.0)	3.4 (1.2, 9.1)	3.3 (1.1, 8.7)
Sodium, mmol/L	139.0 (135.0, 142.0)	139.0 (135.0, 142.0)	139.0 (135.0, 142.0)
Potassium, mmol/L	4.1 (3.6, 4.6)	4.1 (3.6, 4.6)	4.1 (3.6, 4.6)
Chloride, mmol/L	103.0 (98.0, 106.0)	103.0 (98.0, 106.0)	103.0 (98.0, 106.0)
Calcium, mg/dL	8.5 (8.0, 9.0)	8.5 (8.0, 9.0)	8.5 (8.0, 9.0)
Phosphate, mg/dL	3.3 (2.6, 4.0)	3.3 (2.6, 4.0)	3.3 (2.7, 4.1)
Bicarbonate, mmol/L	23.7 (19.3, 28.0)	23.7 (19.3, 28.0)	23.8 (19.4, 28.0)
INR	1.1 (1.0, 1.2)	1.1 (1.0, 1.2)	1.1 (1.0, 1.2)
aPTT, seconds	29.9 (27.1, 34.0)	29.9 (27.2, 34.2)	29.9 (27.1, 33.8)
D-dimer, ug/mL	3.6 (1.6, 8.1)	3.6 (1.5, 7.7)	3.9 (1.8, 9.3)
Lactate dehydrogenase, U/L	253.0 (196.0, 361.0)	252.0 (196.0, 361.0)	255.0 (197.0, 361.0)
NT-pro-BNP, pg/mL	3146.0 (836.5, 11,617.0)	3142.0 (823.8, 11,648.5)	3185.0 (856.8, 11,580.8)
Total bilirubin, mg/dL	0.6 (0.4, 1.1)	0.6 (0.4, 1.1)	0.6 (0.4, 1.1)
Alanine transaminase, U/L	25.0 (15.0, 44.0)	25.0 (15.0, 45.0)	25.0 (15.0, 44.0)
Aspartate transaminase, U/L	29.0 (20.0, 51.0)	29.0 (20.0, 51.0)	29.0 (20.0, 50.0)
Alkaline phosphatase, U/L	95.0 (70.0, 147.0)	95.0 (69.0, 147.0)	94.0 (70.0, 147.0)
Gamma-glutamyl transferase, U/L	54.0 (25.0, 125.0)	53.0 (25.0, 125.0)	54.0 (24.0, 126.0)
UPCR, mg/mg	0.43 (0.13, 1.72)	0.44 (0.13, 1.73)	0.40 (0.12, 1.67)
Concomitant Medications			
Calcium channel blockers, *n* (%)	6412 (55.0)	4517 (55.3)	1895 (54.2)
Beta-blockers, *n* (%)	5164 (44.3)	3636 (44.5)	1528 (43.7)
Alpha-blockers, *n* (%)	3672 (31.5)	2592 (31.8)	1080 (30.9)
RAS inhibitors, *n* (%)	5710 (49.0)	3969 (48.6)	1741 (49.8)
Anti-platelets, *n* (%)	4472 (38.4)	3154 (38.6)	1318 (37.7)
Nitrates, *n* (%)	3195 (27.4)	2236 (27.4)	959 (27.4)
Warfarin, *n* (%)	758 (6.5)	538 (6.6)	220 (6.3)
Statins, *n* (%)	2903 (24.9)	2028 (24.8)	875 (25.0)
Diuretics, *n* (%)	2414 (20.7)	1690 (20.7)	724 (20.7)
NSAID, *n* (%)	5550 (47.6)	3885 (47.6)	1665 (47.6)
COX-2 inhibitors, *n* (%)	1633 (14.0)	1143 (14.0)	490 (14.0)
Metformin, *n* (%)	1703 (14.6)	1192 (14.6)	511 (14.6)
Sulfonylurea, *n* (%)	1085 (9.3)	760 (9.3)	325 (9.3)
Meglitinide analogues, *n* (%)	1050 (9.0)	735 (9.0)	315 (9.0)
SGLT2 inhibitors, *n* (%)	47 (0.4)	33 (0.4)	14 (0.4)
Dipeptidyl peptidase-4 inhibitors, *n* (%)	1330 (11.4)	931 (11.4)	399 (11.4)
Insulin, *n* (%)	5543 (47.5)	3895 (47.7)	1648 (47.1)

Data are presented as *n* (%) or median and interquartile range. *—calculated by the chronic kidney disease epidemiology collaboration (CKD-EPI) creatinine equation. Abbreviations: ICU—intensive care unit; LDL-C—low-density lipoprotein cholesterol; HDL-C—high-density lipoprotein cholesterol; HbA_1c_—glycated hemoglobin; eGFR—estimated glomerular filtration rate; INR—international normalized ratio; NT-pro-BNP—N-terminal pro-brain natriuretic peptide; COPD—chronic obstructive pulmonary disease; HGB—hemoglobin; RAS—renin–angiotensin system; NSAIDs—nonsteroidal anti-inflammatory drugs; COX—cyclooxygenase; SGLT2—sodium–glucose cotransporter 2.

**Table 2 biomedicines-10-00546-t002:** Model performance in predicting risk for end-stage renal disease among the sepsis survivors.

Model	AUC	Accuracy	F1	Precision	Recall	Average Precision	Sensitivity	Specificity
GBDT	0.879	0.891	0.716	0.853	0.617	0.784	0.969	0.617
LGBM	0.868	0.889	0.712	0.851	0.612	0.782	0.969	0.612
Extra-trees	0.865	0.878	0.661	0.876	0.531	0.754	0.978	0.531
Random forest	0.864	0.860	0.565	0.927	0.406	0.765	0.991	0.406
XGBoost	0.859	0.885	0.708	0.820	0.623	0.769	0.961	0.623
Logistic regression	0.854	0.869	0.665	0.780	0.580	0.733	0.953	0.580

Abbreviations: AUC—area under the curve of receiver operating characteristic curve; GBDT—gradient boosting decision tree; LGBM—light gradient boosting machine; XGBoost—extreme gradient boosting.

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
