# Peer review of "Artificial Intelligence for Risk Prediction of End-Stage Renal Disease in Sepsis Survivors with Chronic Kidney Disease"

_biomedicines, 2022, doi:10.3390/biomedicines10030546_

Round 1

Reviewer 1 Report

The manuscript entitled Artificial Intelligence for Risk Prediction of End-stage Renal Disease in Sepsis Survivors With Chronic Kidney Disease addresses a very important topic with an interesting idea in improving the screening of patients with preexisting chronic kidney disease for progression to ESRD after sepsis using artificial intelligence.

Although this is a single-center study on a large number of patients, there are data discrepancies in the manuscript, the results are incomplete and discussion is insufficient. 

Below are my comments and suggestions to the authors.

1) Data collection and study population

Data collection period is unclear – please see below

In Materials and Methods Line 70 - "data from 2010 to 2018"

In Results section Study population line 125 – “A total of 11,661 sepsis survivors from 2001 to 2008 were included in our final…“

In Discussion line 196 – “…during an 8-year follow-up...“

Why are patients only 20 years of age and older included in the study, and most other studies include people 18 years of age and older?

2) Selection of the equation for estimating GFR

Materials and Methods Line 101-102  "The eGFR was estimated using by Modification of Diet in Renal Disease equation. " Why using MDRD instead of CKD EPI equation which is recommended by KDIGO, NKF etc.?

CKD is defined as abnormalities of kidney structure or function, present for ≥3 months, with implications for health. Criteria are either decreased eGFR below 60 mL/min/1,73 m2 and/or present marker(s) of kidney damage (albuminuria, urine abnormalities, structural abnormalities on imaging etc..). MDRD equation is less accurate in estimating GFR above 60 mL/min compared to CKD EPI, and GFR estimated by MDRD at values above 60 mL/min should be expressed only as those above 60, not as absolute value. Also, MDRD underestimates GFR. Moreover, patients with CKD are also patients with GFR 60-90 or normal above 90 mL/min with other aforementioned abnormalities. Therefore, the selection of the equation for estimating GFR in this study is inadequate because patients with CKD may have GFR 60-90 or normal with other pathological markers, and I assume that such were included under the diagnosis of CKD in this study. However, we cannot know for sure considering considering the method of data collection.

3) The data of a research that is considered important and is discussed in the Discussion should be listed in the Results (textual or tabular or graphical...)

Examples from the manuscript

Discussion Line 213-215 “Our results revealed lower long-term mortality and re-hospitalization rates in sepsis survivors than those in previous studies, which might be attributed to differences in disease severity.“

Discussion Line 222-223 “Of the patients, 1367 (11.7%) progressed to ESRD during the follow-up period, with an incidence of 5.1%. “

These data are not mentioned anywhere in the Results, and they are mentioned and compared with the results from other studies in the Discussion.

4) In Table 1, the units of measurement are misspelled: g/Dl, mg/Dl, ug/Ml.. Dl should be dL, Ml should be mL...

In Table 1, the meaning of the HGB abbreviation is not stated.

In Table 1, it should be stated which equation was used to estimate the GFR.

5) In the Discussion, the authors should certainly compare their ESRD progression risk assessment models with existing ESRD progression risk assessment calculators in 2 and 5 year periods (e.g. kidneyfailurerisk.com). What makes this model better, and what are the potential pitfalls?

6) The GBDT model revealed the strong effect of low estimated glomerular filtration rates at discharge in predicting future ESRD.  – this was mentioned both in Abstract and Discussion. What is low eGFR? eGFR lower than 25 mL / min?

Reviewer 2 Report

I read with interest the paper by Lee et al. about the use of AI in sepsis survivors. Some minor points could be discussed to improve the quality of the paper.

  • The adopted model poses low eGFR as the principal determinant of ESRD, followed by low Hb and proteinuria. Because these three parameters are interconnected, could the Authors better outline if the model is able to discriminate the independence of each variable?
  • If I correctly understand data reported in tables, about half of the cohort received RAS inhibitor, which is known as protective on ESRD develpement; however, the possible “protective” effect is not investigated or shown in the model application. Please, discuss and expand model application/text/tables according to this suggestion
  • For the same question, hypertension at discharge (another critical factor for ESRD progression) is not clearly investigated: hypertensive patients are subjects with at least one anti-hypertensive drug? HTN is considered in the analysis and included in the caption of figure 3 (Feature importance plot and SHAP summary plot of the top 20 clinical features predict end-stage renal disease 175 development in the GBDT model) but not effectively reported in the two panels. Please, include the hypertensive definition in methods, reconcile and discuss.

Round 2

Reviewer 1 Report

Major revision of the manuscript has been done following the comments and suggestions of the reviewers. Minor comment and suggestion is to omit % after 70 in line 112 "a validation dataset with a ratio of 70%:30%". 

Author Response

Thanks for the reviewer's comment. We have omitted the % after the ratio numbers. Please see the revised manuscript with tracked changes.